



# What was the source of the atmospheric $CO_2$ increase during the Holocene?

Victor Brovkin[1], Stephan Lorenz[1], Thomas Raddatz[1], Tatiana Ilyina[1], Irene Stemmler[1], Matthew
Toohey[2], and Martin Claussen[1,3]

[1]Max-Planck Institute for Meteorology, Hamburg, Germany
[2]GEOMAR Helmholtz Centre for Ocean Research Kiel, Germany
[3]Meteorological Institute, University of Hamburg, Germany

*Correspondence to*: Victor Brovkin (victor.brovkin@mpimet.mpg.de)

**Abstract.** The atmospheric $CO_2$ concentration increased by about 20 ppm from 6000 BCE to pre-industrial (1850 CE).
Several hypotheses have been proposed to explain mechanisms of this $CO_2$ growth based on either ocean or land carbon
sources. Here, we apply the Earth System model MPI-ESM-LR for two transient simulations of climate and carbon cycle
dynamics during this period. In the 1st simulation, atmospheric $CO_2$ is prescribed following ice-core $CO_2$ data. In response to
the growing atmospheric $CO_2$ concentration, land carbon storage increases until 2000 BCE, stagnates afterwards, and
decreases from 1 CE, while the ocean continuously takes $CO_2$ out of atmosphere after 4000 BCE. This leads to a missing
source of 166 Pg of carbon in the ocean-land-atmosphere system by the end of the simulation. In the 2nd experiment, we
applied a $CO_2$-nudging technique using surface alkalinity forcing to follow the reconstructed $CO_2$ concentration while
keeping the carbon cycle interactive. In that case the ocean is a source of $CO_2$ from 6000 to 2000 BCE due to a decrease in
the surface ocean alkalinity. In the prescribed $CO_2$ simulation, surface alkalinity declines as well. However, it is not
sufficient to turn the ocean into a $CO_2$ source. The carbonate ion concentration in the deep Atlantic decreases in both the
prescribed and the interactive $CO_2$ simulations, while the magnitude of the decrease in the prescribed $CO_2$ experiment is
underestimated in comparison with available proxies. As the land serves as a carbon sink until 2000 BCE due to natural
carbon cycle processes in both experiments, the missing source of carbon for land and atmosphere can only be attributed to
the ocean. Within our model framework, an additional mechanism, such as surface alkalinity decrease, for example due to
unaccounted carbonate accumulation processes on shelves, is required for consistency with ice-core $CO_2$ data. Consequently,
our simulations support the hypothesis that the ocean was a source of $CO_2$ until the late Holocene when anthropogenic $CO_2$
sources started to affect atmospheric $CO_2$.



## 1. Introduction

The recent interglacial period, the Holocene, began about 9700 BCE (Before Common Era), and is characterized by a relatively stable climate. In geological archives, the Holocene is the best-recorded period, making it possible to reconstruct changes in climate and vegetation in remarkable detail (e.g., Wanner et al., 2008). Proxy-based reconstructions suggest a decrease in sea surface temperatures in the North Atlantic (Marcott et al., 2013; Kim et al., 2004) simultaneous with an increase in land temperature in western Eurasia (Baker et al., 2017), so that net changes in the global temperature are small. From Antarctic ice-core records, we know that the atmospheric $CO_2$ concentration increased by about 20 ppm between 5000 BCE and the pre-industrial period (Monnin et al., 2004; Schmitt et al., 2012; Schneider et al., 2013). Hypotheses explaining $CO_2$ growth in the Holocene could be roughly subdivided into ocean- and land-based. The ocean mechanisms include changes in carbonate chemistry as a result of carbonate compensation to deglaciation processes (Broecker et al., 1999; Broecker et al., 2001; Joos et al., 2004), redistribution of carbonate sedimentation from deep ocean to shelves, mostly due to coral reef regrowth (Ridgwell et al., 2003; Kleinen et al., 2016), $CO_2$ degassing due to increase in sea surface temperatures, predominantly in tropics (Indermühle et al., 1999; Brovkin et al., 2008), and decrease in marine soft tissue pump in response to circulation changes (Goodwin et al., 2011). Recent synthesis of carbon burial in the ocean during last glacial cycle suggests excessive accumulation of $CaCO_3$ and organic carbon in the ocean sediments during deglaciation and Holocene (Cartapanis et al., 2018), implicitly supporting the ocean-based mechanism of atmospheric $CO_2$ growth in response to decrease in the ocean alkalinity.

The land-based explanations suggest reduction in natural vegetation cover, such as decreased boreal forests area and increased desert in North Africa (e.g., Foley, 1994; Brovkin et al., 2002), and anthropogenic land cover changes, mainly deforestation (Ruddiman, 2003, 2017; Kaplan et al., 2011; Stocker et al., 2014; Stocker et al., 2017). A couple of land processes ($CO_2$ fertilization, peat accumulation) should have led to terrestrial carbon increase in the Holocene (Yu, 2012; Stocker et al., 2017), complicating the land source explanation. The land-based hypotheses, as well as the ocean soft tissue pump explanation, also have difficulty to conform with the atmospheric $\delta^{13}CO_2$ reconstructions that show no substantial changes in the Holocene (Schmitt et al., 2012; Schneider et al., 2013), contrary to expectation of significant atmospheric $\delta^{13}CO_2$ decrease due to release of isotopically-light biological carbon. For detailed overview of process-based hypotheses, see Brovkin et al. (2016).

While many numerical experiments have been done to test above-mentioned hypotheses with intermediate complexity models, this class of models is limited in spatial and temporal resolution, and consequently does not resolve well climate patterns and variability. Here, we apply the full-scale Earth System model MPI-ESM-LR for simulations of the coupled climate and carbon cycle during the period from 6000 BCE to 1850 CE. The focus of this paper is on the carbon cycle



dynamics for terrestrial and marine components while changes in climate are considered in the companion paper by Bader et al. (submitted).

## 2. Methods

The MPI-ESM-1.2LR used in the Holocene simulations consists of the coupled general circulation models for the atmosphere and the ocean, ECHAM6 (Stevens et al., 2013) and MPIOM (Jungclaus et al., 2013), respectively, the land surface model JSBACH (Raddatz et al., 2007; Reick et al., 2013) and the marine biogeochemistry model HAMOCC5 (Ilyina et al., 2013). In comparison to the MPI-ESM-LR model used in the Climate Model Intercomparison Project, phase 5 (CMIP5) simulations (Giorgetta et al., 2013), the model has been updated with several new components for land hydrology and carbon cycle. In addition to the previously developed dynamic vegetation model (Brovkin et al., 2009), the new JSBACH component includes the soil carbon model YASSO (Goll et al., 2015), a 5-layer hydrology scheme (Hagemann and Stacke, 2015) and an interactive albedo scheme (Vamborg et al., 2011). HAMOCC was updated with prognostic nitrogen fixers (Paulsen et al., 2017), and a parameterization for a depth-dependent detritus settling velocity and remineralization of organic material (Martin et al., 1987).

We used a combination of several forcings as boundary conditions for transient simulations (Fig. 1). The orbital forcing (Fig. 1a) follows a reconstruction by Berger (1978). We use a solar irradiance forcing that was reconstructed from $^{14}$C tree-ring data that correlates with the past changes in the solar open magnetic field (Krivova et al., 2011). $CO_2$, $N_2O$ and $CH_4$ forcings stem from ice-core reconstructions (F. Joos, personal communication, Fig. 1b,c). We applied a new reconstruction of volcanic forcing (Bader et al., submitted) based on the GISP2 ice core volcanic sulphate record (Zielinski et al., 1996) (Fig. 1d). To mantain consistency with the CMIP5 simulations, we included the landuse changes based on the LUH1.0 product by Hurrt et al. (2011) and Pongratz et al. (2011), provided for the period after 850 CE. To avoid an abrupt change in the landuse forcing at 850 CE, we interpolated the landuse map from 850 CE linearly backwards to no-landuse conditions at 150 BCE (Fig. 1e). The landuse areas in 1750 CE are in line with the most recent update of the HYDE dataset (Goldewijk et al., 2017), although our interpolation underestimates crop and pasture areas earlier than 1750 CE.

The spinup simulation 8KAF started from initial conditions for pre-industrial climate and continued with boundary conditions for 6000 BCE for 1,000 years in order to establish an equilibrium of climate and carbon cycle with the boundary conditions. During the spinup period the atmospheric $CO_2$ concentration was kept at a constant level of 260 ppm. Afterwards, the model was run with an interactive carbon cycle for 100 years to ensure a dynamic equilibrium between land, ocean, and atmospheric carbon cycle components (simulation 8KAFc).





We performed two transient simulations from 6000 BCE to 1850 CE, a commonly-defined onset of the industrial period. The first transient simulation, TRAF, was initiated from the end of the spinup simulation 8KAF. In the TRAF simulation, the atmospheric $CO_2$ concentration is prescribed from ice core reconstructions. Consequently, land and ocean carbon uptakes do not interfere with each other and the total mass of carbon in the land-ocean-atmosphere system is not conserved.

The 2nd transient simulation, TRAFc, started from the end of the 8KAFc simulation in the interactive climate-carbon cycle mode. Using equilibrium initial conditions and boundary conditions for the carbon cycle does not guarantee that the interactively simulated atmospheric $CO_2$ concentration will follow the trend reconstructed from ice-cores. If the simulated atmospheric $CO_2$ concentration differs substantially from reconstructed data, both climate and carbon cycle components

deviate from results which would be obtained if the model was driven by the reconstructed $CO_2$ forcing, and these biases complicate the comparison of trends between the model and observations. To ensure that simulated $CO_2$ is close to the reconstructed time series, in the TRAFc simulation we used a $CO_2$-nudging technique following the approach of Gonzales and Ilyina (2016), targeting the atmospheric $CO_2$ record from ice-core reconstructions. If the simulated atmospheric $CO_2$ dropped below the target, the surface ocean total alkalinity and dissolved inorganic carbon (DIC) concentrations were

decreased in a 2:1 ratio to mimic the process of $CaCO_3$ sedimentation. If $CO_2$ was higher than the target, the alkalinity and DIC were not changed. This alkalinity-forced approach is supported by evidence of decreasing deep ocean carbonate ion concentration in the course of the Holocene (Yu et al., 2014) and excessive coral reef buildup on shelves during deglaciation (Opdyke and Walker, 1992; Vecsei and Berger, 2004) as well as recent synthesis by Cartapanis et al. (2018). The alkalinity decline resembles excessive shallow-water carbonate sedimentation such as coral reefs, which are not included in

HAMOCC.

HAMOCC includes a module of sediment processes (Heinze et al., 1999). Interactive simulation of the sediment pore water chemistry and accumulation of solid sediment components, such as $CaCO_3$, particulate organic carbon (POC), opal, and clay is a necessary condition to calculate changes in the ocean biogeochemistry on millennial timescales. To compensate for the

POC, $CaCO_3$, and opal losses due to sedimentation, the fluxes into the sediment over the last 300 years of the spin-up runs were analysed, and a globally uniform weathering input for silicate, alkalinity, nutrients in the form of organic matter, and dissolved inorganic carbon was prescribed. Note that the weathering flux calculated using this approach is sensitive to changes in the model setups (prescribed versus interactive $CO_2$). This explains the difference between weathering fluxes in TRAF and TRAFc experiments (Table 1).






## 3. Results and Discussion

### 3.1 Global Response

The setup of the TRAF simulation resembles experiments performed in CMIP5 with $CO_2$ concentrations prescribed from Representative Concentration Pathway (RCP) scenarios. Similar to the RCP simulations, changes in carbon pools on land

and in the ocean could be estimated from land-atmosphere and ocean-atmosphere fluxes, respectively. Cumulative changes in the ocean and land $CO_2$ fluxes reveal that, by the end of the simulation, the ocean is a sink of 152 PgC, while the land is a source of 39 PgC (Fig. 2a, Table 1). Accounting for an increase in the atmospheric carbon pool by 53 PgC, the total carbon budget has a deficit of 166 PgC by 1850 BCE. Assuming that $CO_2$ airborne fraction on millennial timescale is about 1/6 (Maier-Reimer and Hasselmann, 1987) or even less if we account for the land response due to $CO_2$ fertilization, the

atmospheric $CO_2$ concentration by the end of the simulation would be by 11-13 ppm less than observed (286 ppm). Therefore, carbon budget changes in the TRAF experiment imply that other boundary conditions are necessary to obtain the amplitude of the simulated $CO_2$ concentration trend as reconstructed from the ice cores.

Carbon cycle changes in the interactive $CO_2$ simulation TRAFc are shown in Fig. 2b. At the beginning of the simulation

(during 6 to 5000 BCE), atmospheric $CO_2$ and land carbon fluctuate with an amplitude of several PgC, while the ocean becomes a small source of $CO_2$ to the atmosphere. Between 5000 and 2000 BCE, atmospheric carbon storage increases by about 30 PgC, the land takes about 60 PgC due to $CO_2$ fertilization, while ocean releases ca. 90 PgC. Between 2000 BCE and 1 CE (start of the Common Era; note that year 0 does not exist in the CE system), land is a source of 10 PgC to the atmosphere. After 1 CE, land carbon losses accelerate due to land-use changes, and by 1850 CE land carbon decreases by an

additional 60 PgC. In 1850 CE, land and ocean are sources of 17 and 39 PgC, respectively, while the atmosphere gains 56 PgC. The atmospheric minimum in $CO_2$ around 1600 CE apparent in the reconstruction is not reproduced by the model, confirming that the abrupt uptake of carbon by land or ocean is difficult to attribute to internal variability in the coupled climate-carbon system (Pongratz et al., 2011) and external forcing scenarios, such as an abrupt reforestation of tropical America (Kaplan et al., 2011), might need to be accounted for.

Decadal-scale excursions in ocean and land carbon storages in Fig. 2b are mainly explained by responses to surface cooling resulting from volcanic eruptions. The most visible example of this $CO_2$ response is during the period around 3200 BCE, when reconstructed aerosol optical depth shows an enhancement which is moderate in magnitude but of long duration (Fig. 1d), potentially resulting from a long-duration high latitude eruption, or from contamination of the volcanic record by

biogenic sulphate (Zielinski et al., 1994). In response to this applied forcing, the land takes up carbon due to decreased respiration (see, e.g., Brovkin et al., 2010; Segschneider et al., 2013), while the alkalinity adjustment in the ocean counteracts the land carbon uptake, leading to carbon release from the ocean. After a few decades, the land turns into a source of carbon due to reduced productivity, ocean carbon uptake restores, and atmospheric $CO_2$ reveals a spike due to an




excessive land source. At 3000 BCE, this spike ceases and the simulated $CO_2$ continues to fluctuate around the ice-core time series. Although the volcanic forcing included in the simulations at 3200 BCE is likely an overestimate, this case illustrates the response of the climate-carbon system to an extreme volcanic aerosol forcing which leads to pronounced cooling of the land and ocean surfaces.

Changes in carbon density on land and in the ocean in the course of both TRAF (not shown) and TRAFc simulations reveal complex patterns (Fig. 3). In the ocean, the vertically-integrated DIC is decreasing everywhere, causing negative change patterns dominating in the Southern Ocean and Northern Pacific. Carbon sedimentation is high in upwelling zones, mainly in coastal areas and the tropical Pacific, and that causes strong accumulation patterns. The land has a mixed pattern of increased

carbon density, mostly in South America and in central North America, with decreased densities in Africa and East Asia. This is caused by an interplay between climate, $CO_2$, and land use effects on soil and biomass storages.

Changes in the carbon budget components over the experimental period are provided in Table 1. For the atmosphere, the difference between the TRAF and TRAFc simulations is minor (3 PgC). For the land, a difference of 22 PgC is caused

mainly by the relatively higher $CO_2$ concentration in the TRAFc simulation, especially during the period of lower $CO_2$ around 1600 CE, due to the $CO_2$-fertilization effect on the plant productivity. The ocean-to-atmosphere cumulative fluxes (-152 and 39 PgC for TRAF and TRAFc, respectively) are minor in comparison with the ocean carbon budget components, and the difference of 191 PgC is explained by the applied surface alkalinity removal in the TRAFc simulation. The carbon inventory of the water column that predominantly includes dissolved inorganic carbon (DIC) loses 1324 and 1799 PgC in the

TRAF and TRAFc runs, respectively. Sediments accumulate more than 3,500 PgC in the form of $CaCO_3$ and organic carbon, mainly compensated by the weathering flux from land. In the TRAFc experiment, 1,224 PgC was removed from the ocean surface in the form of $CaCO_3$, effectively reducing the weathering flux (3,270 PgC) to a scale below the TRAF experiment (2,137 PgC). In total, despite large changes in the cumulative fluxes of weathering and sedimentation, the net cumulative ocean-to-atmosphere flux is minor.

**3.2 Land carbon and vegetation**

Natural changes in vegetation and tree cover are most pronounced for the time slice around 1 CE, before the start of substantial landuse forcing. Comparing with 6000 BCE, vegetation cover becomes much less dense in Africa, mainly due to decreased rainfall in response to the decreasing summer radiation in the Northern Hemisphere (Fig. 4a). Boreal forests moved southward in both North America and Eurasia (Fig. 4b). The southward shift of vegetation in North Africa, and of the

treeline in Eurasia from mid-Holocene to pre-industrial, as well as the increase in vegetation and tree cover in central North America, is in line with pollen evidence (Prentice et al., 2000). The southward retreat of the boreal forest in North America is much less pronounced than in Eurasia (Fig. 4b). This is also in line with reconstructions, as there is no evidence for a



significant shift of the treeline in North America (Bigelow et al., 2003), likely due to the cooling effects of the remains of the Laurentide ice sheet, which is not accounted as a forcing in our simulations.

Simulated changes in vegetation cover are reflected in the carbon density changes (Fig. 5a). From 6000 BCE to 1 CE, the
carbon density decreases in Northern Africa, East Asia, northern South America and above 60°N slightly in Eurasia. In most of the rest of land ecosystems, the carbon density increases, mostly due to $CO_2$-fertilization effects as the atmospheric $CO_2$ concentration increases by about 15 ppm by 1 CE. A strong increase in the Southern Hemisphere and Central North America is also due to increased vegetation density. After 1 CE, land carbon declines due to landuse changes, predominantly deforestation (Fig. 5b). Patterns of carbon decrease after 1 CE reflect landuse patterns except in South America, South
Africa, and central North America. The simulated increase in land carbon storage before 2000 CE and decrease afterwards is consistent with the changes in atmospheric $\delta^{13}CO_2$ (Schmitt et al., 2012).

### 3.3 Ocean carbon

Simulated physical ocean fields, including sea surface temperatures and the Atlantic meridional overturning, do not change substantially in the Holocene. The main reason for the declining carbon storage in the water column (Fig. 3, Table 1) is a decrease in ocean alkalinity (Fig. 6a; Fig. 7a). This is explained by the applied surface ocean alkalinity forcing and also by a response of the ocean carbonate chemistry to changes in carbonate production. The global $CaCO_3$ export from surface to aphotic layer increases by about 5% between 6000 and 2000 BCE in both TRAF and TRAFc simulations and returns to the
6000 BCE level by the end of the simulation. Comparing TRAF and TRAFc simulations, the difference in the globally-averaged ocean alkalinity in these two simulations by 1850 CE is 35 μmol/kg, similar to the difference in surface alkalinity changes shown on Fig. 6a. Accounting for 7850 years of experimental length, the required excessive carbonate sedimentation in the shallow waters would be 3 Tmol/yr or at the lower bound of estimates of 3.35 to 12 Tmol/yr $CaCO_3$ accumulation proposed by Vecsei and Berger (2004) and Opdyke and Walker (1992).

Besides the estimate of applied carbonate accumulation forcing, another way to address the plausibility of simulated alkalinity trends is to compare changes in the carbonate ion concentration ($[CO_3^=]$) in the deep Atlantic and Pacific oceans with available reconstructions of carbonate ion concentrations. Using the benthic foraminiferal B/Ca proxy for deep water $[CO_3^=]$, Yu et al. (2014) found that $[CO_3^=]$ in the deep Indian and Pacific Oceans declined by 5–15 μmol kg$^{-1}$ during the
Holocene. Broecker et al. (1999) and Broecker & Clark (2007) suggested a similar amplitude of $[CO_3^=]$ changes in the deep Atlantic. Comparison of changes in $[CO_3^=]$ in TRAF and TRAFc simulations with $[CO_3^=]$ data reconstructed by Yu et al. (2013) reveals a significant difference between TRAF and TRAFc in the Atlantic (Fig. 6b). Decrease in $[CO_3^=]$ in the TRAFc simulation is more significant than in the TRAF experiment, presumably due to a stronger decrease in alkalinity in



the former simulation. Interestingly, changes in [$CO_3^=$] at the Pacific site are not significant in both simulations, while the data propose a slight decrease in carbonate ion concentration. The difference between the Atlantic and Pacific responses is visible in Fig. 7. In both experiments, simulated changes in [$CO_3^=$] in the Atlantic and Southern Oceans are stronger than in the Indo-Pacific. At a depth of 4 km, comparable with the depth of the cores by Broecker & Clark (2007), changes in the

tropical oceans in both simulations are in the range of 0-15 mol m$^{-3}$. Changes in the TRAFc experiment are more pronounced than in TRAF due to stronger changes in alkalinity. As expected, [$CO_3^=$] changes are more pronounced for depths of 3 km than for 4 km (Fig. 7).

Comparison of simulated ocean carbon budget with recent carbon data synthesis (Cartapanis et al., 2018) is shown in the

Table 2. In comparison with mean data values, $CaCO_3$ burial in the model (27.9-29.1, plus 13 TmolC/yr surface removal in the TRAFc experiment) is higher than in the data (23.3 TmolC/yr), however, this is compensated by higher modelling weathering rate (24.6-34.7 TmolC/yr) comparing to 11.7 TmolC/yr in the data. The model values are at the upper end of the data uncertainty range (11 – 38 TmolC/yr, see min-max range for $CaCO_3$ sedimentation in the Table 2). Organic carbon burial in the model (10.5-11.3 Tmol/yr) is less than in the averaged data (18.3 Tmol/yr), however, the uncertainty in the

burial is so high (6-58 Tmol/yr) that the model values are almost twice more than the lower end of the data range.

An important question is whether the ocean on average lost carbon content over the Holocene. In equilibrium, volcanic $CO_2$ outgassing (not accounted explicitly in the model), both aerial and submarine, compensates for weathering, therefore for proper comparison it needs to be included into the table as the ocean-atmosphere budget. In that case, averaged ocean water

column losses in the data is 20.8 TmolC/yr.  Similar to the data, the model shows the loss of carbon from the water column, 13.8 and 18.7 TmolC/yr in the TRAF and TRAFc experiments, respectively. Therefore, simulated ocean carbon losses are qualitatively (and even quantitatively for TRAFc) in line with observations.

### 3.4 Limitation of the model setup

There are certain limitations of the carbon cycle models used in the study. Firstly, the applied version of JSBACH does not

include wetland and peatland processes. If the Holocene peat accumulation of several hundred GtC (Yu, 2012) were accounted for, the land would be a stronger sink of carbon during 6000 to 2000 BCE. This might require an even stronger ocean source. On the other hand, we neglect other sources of atmospheric $CO_2$ which might compensate for the peatland growth. Geological sources of methane of the scale of 30-40 Tg/yr are pronounced in intergacials (Bock et al., 2017; Saunois et al., 2016). Although uncertainty in the geological methane source remains high, after oxidation in the atmosphere, this

source would correspond to 200-300 GtC during the last 8,000 years and potentially compensate for a substantial part of the peat growth. Secondly, HAMOCC does not include coral reefs as a process-based component. This is one of the reasons why the surface alkalinity was forced directly in the TRAFc simulation. Thirdly, the applied versions of JSBACH and HAMOCC do not simulate carbon isotope changes, in particular $^{13}C$ changes. For land carbon, the increase in the carbon storage on land



by 50-60 PgC by 2000 BCE and its decrease by 80 PgC by 1850 CE would be translated into a ca. 0.05‰ decrease in atmospheric $\delta^{13}CO_2$. This small change is within the uncertainty bounds of $\delta^{13}CO_2$ reconstructed from ice cores (Schmitt et al., 2012). For ocean carbon, carbonate changes would not significantly modify the ocean and atmospheric $\delta^{13}CO_2$ content. Simulated changes in biological production and export flux might have affected the atmospheric $\delta^{13}CO_2$, but the scale will

likely be small.

In addition, two limitations are intrinsic to the setups of spinup and transient simulations. Firstly, assuming that all carbon cycle components are initially in equilibrium with boundary conditions is a simplification. Changes in climate due to slowly changing boundary conditions, such as orbital or greenhouse gas forcing, are occurring on timescales similar to long-term

processes in the carbon system (soil buildup on land, carbonate compensation in the ocean). Therefore, the carbon cycle is never in full equilibrium, and memory in the carbon cycle processes, due for example to carbonate compensation in the ocean during deglaciation, affects the carbon dynamics afterwards. A proper way to account for the memory effect is to set spin-up simulations millennia before the Holocene, e.g. at the last glacial maximum (19000 BCE), and perform a transient deglaciation simulation. This is presently too challenging for full-scale ESMs due to high computational costs. Secondly, the

weathering fluxes are assumed to be constant during the transient simulation. While this is a common practice for ocean biogeochemistry simulations (e.g., Ilyina et al., 2013; Heinze et al., 2016), it may result in a mismatch between weathering and sedimentation under changing boundary conditions in transient simulations. As land climate evolves, weathering fluxes are changing due to their dependence on runoff and temperature. This causes a shift in land-to-ocean fluxes of carbon, alkalinity and nutrients, leading to inventory changes and a possible drift in ocean-to-atmosphere fluxes. Consequently, to

simulate the carbon budget correctly, models have to include interactive weathering processes. These two caveats (steady-state initial conditions and fixed weathering) apply to both TRAF and TRAFc simulations. On the other hand, simulations with intermediate complexity models suggested that the impact of the memory effect on Holocene carbon dynamics is rather minor (e.g., Menviel and Joos, 2012), and since Holocene climate is quite stable, we do not expect that weathering changes due to minor cooling (warming) in high latitudes (tropics) are substantial.

## 4. Conclusions

Using the Earth System Model MPI-ESM-LR, we performed two transient simulations of the climate and carbon cycle dynamics in the Holocene, one with prescribed atmospheric $CO_2$ and one with interactive $CO_2$ using nudged ocean

alkalinity. In both simulations, the land is a carbon sink during the mid-Holocene (from 6000 to 2000 BCE) and a source of $CO_2$ after 1 CE due to landuse changes. Changes in vegetation cover at 6000 BCE relative to 1 CE (enhanced vegetation cover in North Africa, northward extension of boreal forest in Asia) are in line with available pollen records.




In the prescribed $CO_2$ experiment TRAF, the ocean is a sink of carbon. This strengthens the argument that neither changes in circulation nor in sea surface temperatures are capable of explaining $CO_2$ growth in the Holocene. In the coupled land-ocean-atmosphere system, there is a total deficit of 166 PgC by the end of the experiment. The TRAFc simulation with interactive $CO_2$ is performed in the nudging mode: we use the surface alkalinity changes as the forcing for ocean-atmosphere $CO_2$ flux.

In response to this forcing, the ocean serves as a source of carbon over the Holocene. The alkalinity decline is within the bounds of proposed changes in the carbonate sedimentations in shallow waters and consistent with available proxies for carbonate ion decrease in the deep sea.

There are several limitations of our simulations related to initial conditions and forcings. We cannot simply overcome them

by repeating runs in different setup or by doing additional sensitivity experiments due to the high computational costs of full-scale ESMs. Despite of these limitations, we can make several conclusions on the potential source of $CO_2$ to the atmosphere during the last 8,000 years. Regarding the land source, experiments demonstrate that natural carbon dynamics lead to increase in the land carbon storage during the first half of the simulation (until 2000 BCE). This is in line with previous simulations performed with intermediate complexity models (e.g., Kaplan et al., 2002; Kleinen et al., 2016). During 6000 to

2000 BCE, the atmospheric $CO_2$ increase is about 2/3 of the estimated 20 ppm increase. Although the TRAF and TRAFc simulations do not account for landuse changes during this period, assuming that landuse was a source of carbon to the atmosphere requires about 100 PgC to compensate for natural land, ocean, and atmospheric carbon content increase during this time. If we account for the peat carbon accumulation (neglected in TRAFc), emissions from landuse would need to be higher (about 200 PgC over the period 6000 to 2000 BCE). This is not absolutely impossible (Kaplan et al., 2011), but such a

high-end landuse emission scenario for the end of Neolithic period, when agriculture was not yet widespread in Europe and America, is rather unlikely.

Regarding the ocean source, both TRAF and TRAFc simulations show a decrease in ocean alkalinity. Even if this decrease is a result of a drift in the carbonate system due to imperfect initialization of the balance between sedimentation and

weathering, in both simulations the model is capable of producing a decrease in the carbonate ion concentrations in the Atlantic which is in the direction proposed by proxy data (Fig. 7, b). The magnitude of the decrease in the TRAF experiment is underestimated compared to the proxy data, while it is in line with the data for the TRAFc experiment. As land serves as a carbon sink until 2000 BCE due to natural (non-anthropogenic) carbon cycle processes in both experiments, the missing source of carbon for land and atmosphere could be only attributed to ocean. Within our model framework, an additional

mechanism is required for consistency with ice-core $CO_2$ data, such as surface alkalinity decrease, due for example, to unaccounted carbonate accumulation processes on shelves supported by observational evidence. Finally, our simulations support the hypothesis that the ocean was a source of $CO_2$ until the late Holocene when anthropogenic $CO_2$ sources started to affect atmospheric $CO_2$.



## 5. Code and data availability

The model code is available after request and after acceptance of the MPG license. The data used for the analysis and figures are available from the MPI-M library, contact: publications@mpimet.mpg.de.

## 6. Author contribution

TR and IS contributed to the model development and experimental setup. TI and MC contributed to experimental design of simulations, MT provided volcanic forcing, SL performed the simulations, VB analyzed the simulations and wrote the first draft. All authors contributed to the results discussion and manuscript writing.

## 7. Competing interests

The authors declare that they have no conflict of interests.

## 5. Acknowledgements

This work contributes to the project PalMod funded by the German Federal Ministry of Education and Research (BMBF), Research for Sustainability initiative (FONA, https://www.fona.de). We are grateful to the authors of the companion manuscript by Bader et al. for insightful discussions. We thank Mathias Heinze for helping with the equilibrium model spinup, Veronika Gayler for post-processing the model output, and Estefania Montoya-Duque for archiving the primary data.

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



**Figure captions**

Figure 1. Time series of applied forcings: a) June-July-August insolation at 60°N, Wm$^{-2}$; (b) atmospheric $CO_2$ concentration, ppm, and (c) $N_2O$ and $CH_4$ concentrations, ppb; (d) Aerosol optical depth of volcanic eruptions; (e) global area of crop and pastures, $10^6$ km$^2$.

Figure 2. Changes in the cumulative fluxes for major carbon cycle components (land, ocean, and atmosphere), PgC, from 6000 BCE to 1850 AD in the TRAF (a) and TRAFc (b) simulations. Colour legend: land, cyan; ocean, blue; atmosphere, magenta; ice-core reconstruction – orange.

10    Figure 3. Combined map of changes in the ocean carbon storage (vertically integrated ocean water column plus sediments) and land (soil plus vegetation) at the end of the TRAFc simulation (1850 CE) relative to 6000 BCE, in kgC m$^{-2}$.

Figure 4. Change in vegetation fraction (a) and tree cover fraction (b) at 1 CE relative to 6000 BCE in the TRAFc simulation.

Figure 5. Change in land carbon density, kg C m$^{-2}$, relative to 6000 BCE at 1 CE (a) and at 1850 CE relative to 1 CE (b) in the TRAFc simulation.

Figure 6. (a) 100-yr moving average of global surface alkalinity, μmol kg$^{-1}$. (b) 100-yr moving average of carbonate ion
20    concentration, μmol kg$^{-1}$, averaged for 9 neighbouring model grid cells centred in deep Atlantic (12°N, 60°W, 3400 m) and deep Pacific (1°S, 160°W, 3100 m) The data (circles) are $[CO_3^=]$ for sites VM28-122 and GGC48 reconstructed by Yu et al. (2013) which appropriately correspond to the ocean grid cells accounting for model bathymetry mask. Data uncertainties (1 σ) reported for Atlantic by Yu et al. (2013) are indicated by whiskers. 35‰ salinity is used for model unit conversion from m$^{-3}$ to kg$^{-1}$.

Figure 7. Differences in carbonate ion concentration, mol m$^{-3}$, between TRAFc and TRAF simulations in 1850 CE at the depth of 3 km (a) and 4 km (b).



**Table 1. Changes in compartments and cumulative fluxes at 1850 CE relative to 6000 BCE, PgC**

| Experiment | Atmosphere | Land | Ocean, water | Ocean sediments, $CaCO_3/C_{org}$ | Surface ocean $CaCO_3$ removal | Ocean-to-atmosphere flux | Land-to-ocean flux (weathering[1]) |
|---|---|---|---|---|---|---|---|
| TRAF | 53 | -39 | -1324 | 2628/985 | 0 | -152 | 2137 |
| TRAFc | 56 | -17 | -1799 | 2738/1068 | 1224 | 39 | 3270 |
| TRAFc-TRAF | 3 | 22 | -475 | 110/83 | 1224 | 191 | 1133 |

---

[1] Weathering flux is not accounted in the land compartment changes (second column)





Figure 1





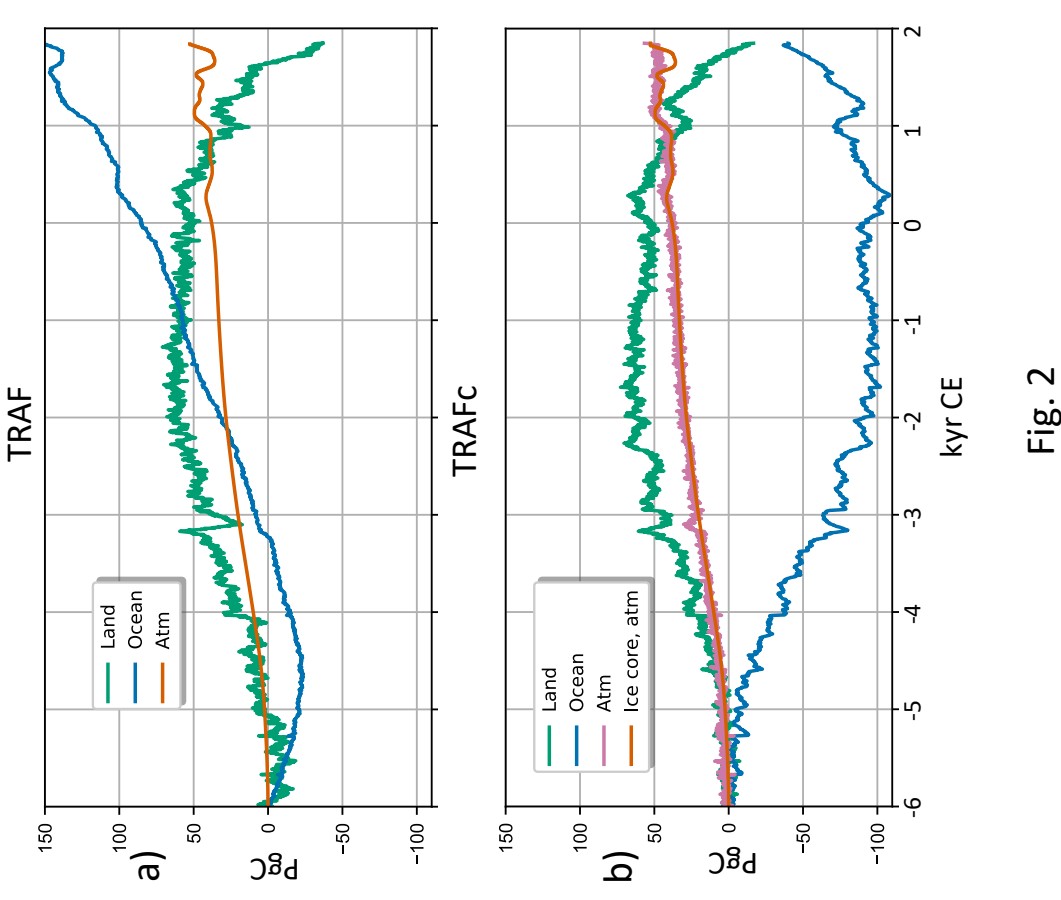





Fig. 3





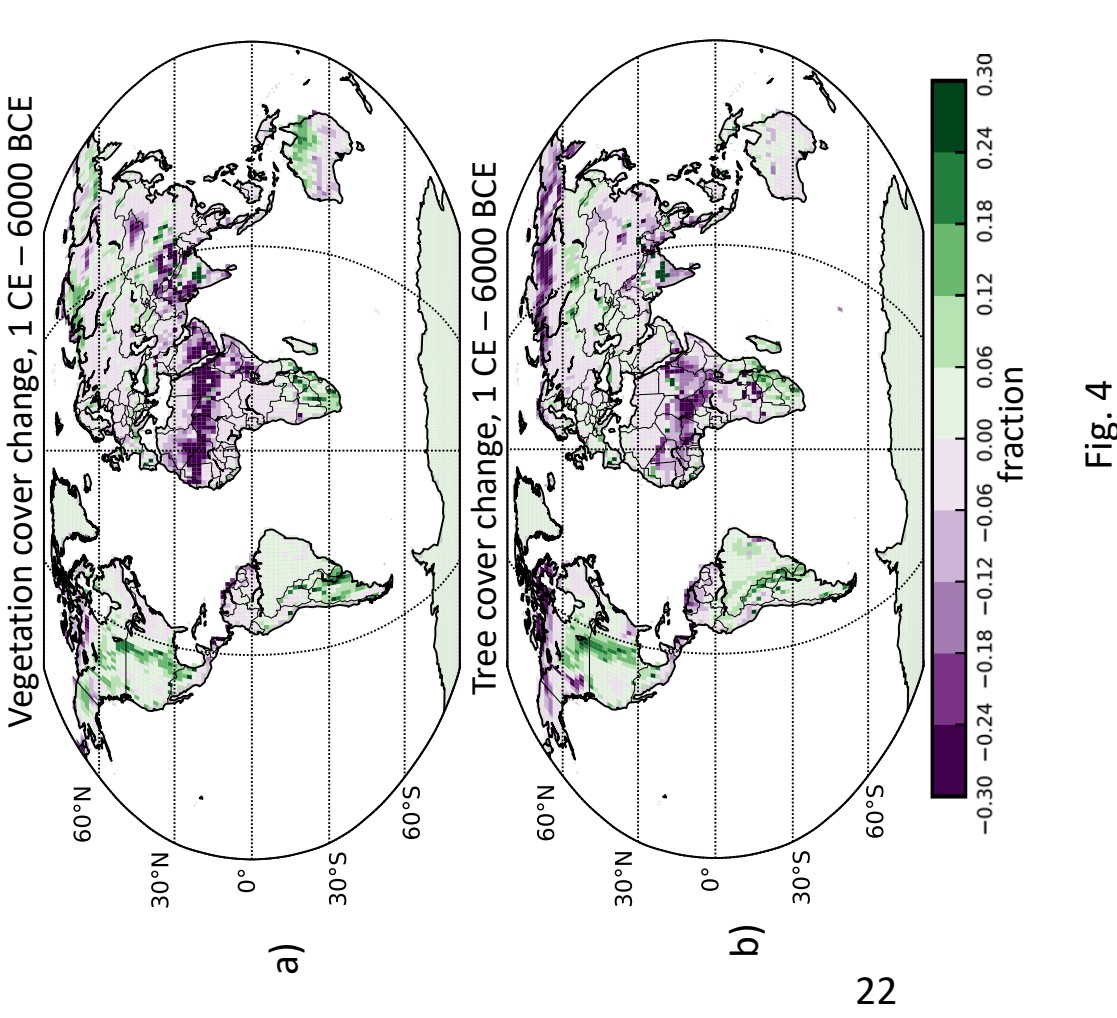

Fig. 4



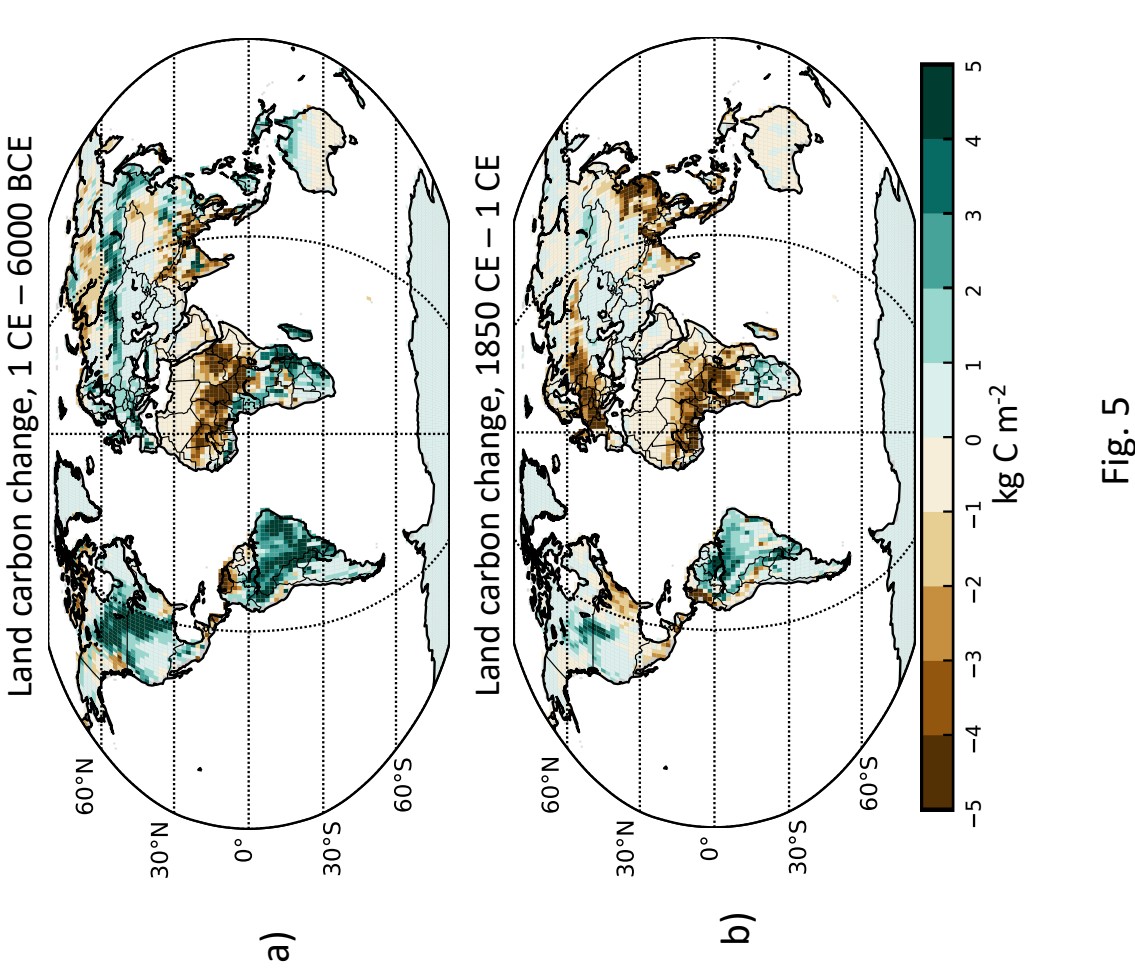

Fig. 5



Fig. 6





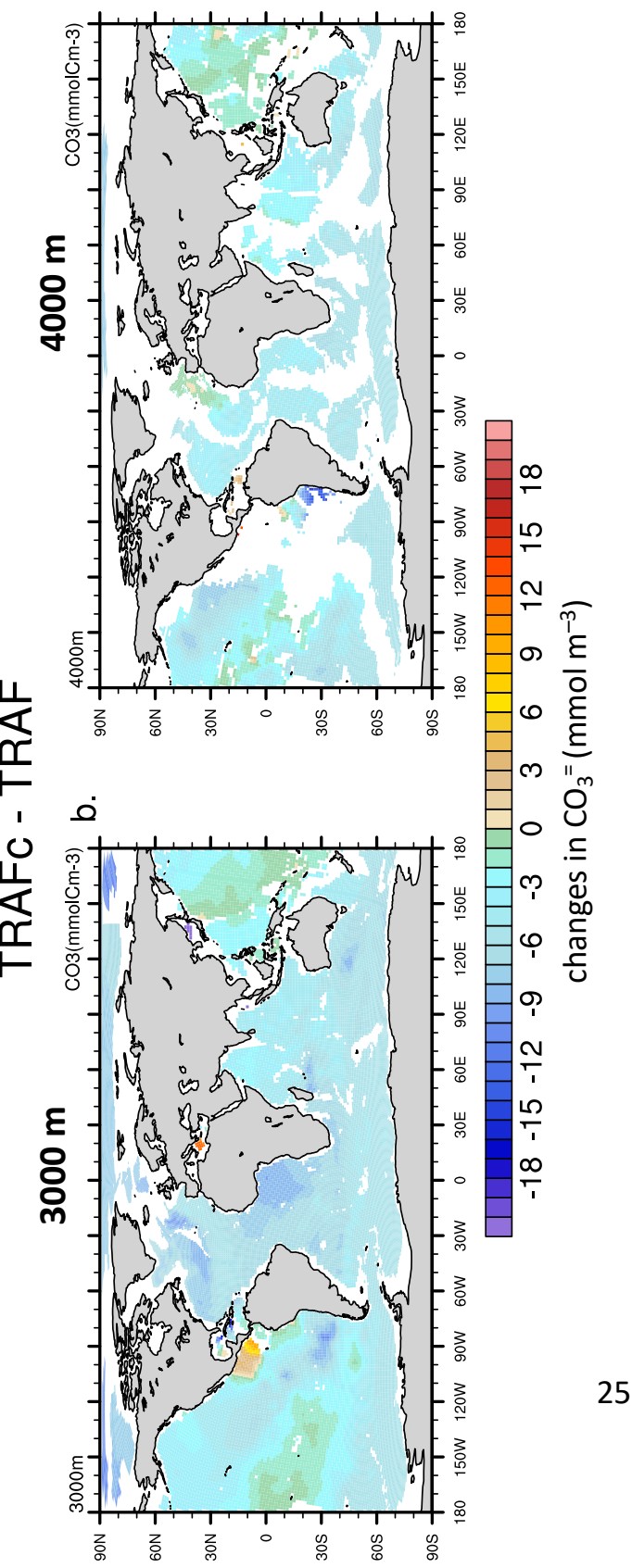

Fig. 7