# Peer review of "What was the source of the atmospheric CO2 increase during the Holocene?"

_Biogeosciences, 2019_

## Referee Comment (RC1) · Anonymous Referee #1 · 9 Apr 2019

Brovkin et al. present the results of two transient simulations covering the period 6000 BCE to pre-industrial 1850 CE with the Earth system model MPI-ESM-LR. The goal is to constrain the processes leading to the changes in atmospheric CO2 concentration during that period. The conclusion of the study is that a surface alkalinity decrease, for example due to enhanced carbonate accumulation on continental shelves, is necessary to explain the Holocene atmospheric CO2 trajectory.

This is an interesting and very valuable study. Please find some comments below that should be addressed before publication.

1) Experimental set up:

The model was equilibrated under 6000 BCE conditions, with an atmospheric CO2

concentration lower (260ppm) than during the pre-industrial. If the global alkalinity concentration was kept constant, then the ocean would have lost carbon during the spin-up phase. Then forced with a transient increase in atmospheric CO2, all else being equal, the ocean should take up carbon. This is apparently what is happening as changes in SST, ocean circulation... are small. Potential bias due to imposing an atmospheric CO2 concentration should be briefly mentioned.

2) Justification of surface alkalinity decrease and comparison with previous studies:

The authors have previously done extensive work on the topic of global carbon cycle changes on glacial-interglacial cycles. They are therefore well aware of the literature on the topic, on the rationale behind decreasing surface alkalinity during the Holocene, and on the results from previous studies. However, given that here it is given as the main mechanism controlling Holocene atmospheric CO2, I would have expected a more in depth introduction of the topic and discussion with respect to previous studies. There is no mention in the introduction of the timing and magnitude of carbonate sedimentation on shelves, as well as on the results of previous modelling studies on the topic. There are only a few words on the topic in the introduction (p2, L.13-14), a few words in the method without any quantification (p4, L.17-18). As a side note, the introduction given in Kleinen, Brovkin et al., (2016) was much more informative. The results and a rapid comparison to Vecsei and Berger (2004) is given p7, L 23-25, but there is no comparison with results from previous studies. In addition, the magnitude of the necessary alkalinity change and its equivalent change in carbonate sedimentation could also be discussed in the context of the simulated changes in land carbon (see section 3).

3) Changes in terrestrial carbon:

Using a mass balance approach and high-resolution atmospheric CO2 and d13CO2 records, Elsig et al., 2009 suggest a land carbon uptake of ∼290 GtC between 11ka and 5 ka B.P. I am surprised to see no reference/discussion to this study. As far as I

can see this result seems supported among others by Stocker et al., (2017), Menviel & Joos (2012)... Here, the model suggests a terrestrial carbon uptake of ∼50 GtC between 8 and 2 ka B.P, which is much smaller and with a different timing. A discussion should be added on the results of this study, compared to the estimates of Elsig et al., (2009). The authors should discuss how their results could be reconciled with the atmospheric d13CO2 record, as also shown in other modelling studies, which included carbon isotopes.

---

## Referee Comment (RC2) · Fortunat Joos (Referee) · 10 Apr 2019

The study by Victor Brovkin and colleagues is interesting. They provide results from first transient fully coupled ESM simulations covering the entire last 8000 years. This is novel and warrants publication.

The conclusion by Brovkin et al. that shallow-water CaCO3 deposition (coral reef growth) plays a role for the late Holocene CO2 increase is similar to the conclusions from earlier studies using EMICs. A difference is that this study seems to imply that shallow water carbonate deposition is by far the most important driver for the late Holocene CO2 increase. This is a possibility, but others found additional factors such as legacy effects of earlier land carbon uptake to be equal or even more important.

[Figure]

Here below my specific comments in addition to those offered by reviewer 1.

1) Information about model drift may be helpful for the reader.

P4, l25-l29: I am puzzled about the large, 50%, difference in the diagnosed weathering flux between the simulations TRAF and TRAFc.

P3, l30: Is the ALK nudging also used during the coupled spin up 8KAFc. If not how large is the drift in CO2?

Both TRAF and TRAFc were first spin up under prescribed CO2 (260 ppm, 8KAF). The spin up is extended by an additional 100 years with an open atmosphere (simulation 8KAFc) before starting TRAFc. The weathering flux is diagnosed from the last 300 yr of the spin up. In other words, the last 200 years of 8KAF are used to diagnose the weathering for TRAF and TRAFc; the difference in the diagnosed weathering for TRAF and TRAFcarises from the other 100 years of results taken either from 8KAF or from 8KAFc.

Why is there such a large difference in the diagnosed weathering flux even though 200 out of 300 years are taken from the same run? Is the model far from equilibrium? Is there a substantial model drift? Is there information from a control run available?

2) The statement on geological methane emissions appears misleading and needs to be revised.

P8, line 28: "Geological sources of methane of the scale of 30-40 Tg/yr are pronounced in intergacials (Bock et al., 2017; Saunois et al., 2016). Although uncertainty in the geological methane source remains high, after oxidation in the atmosphere, this source would correspond to 200-300 GtC during the last 8,000 years and potentially compensate for a substantial part of the peat growth."

The change in geological methane emissions (GEM) over glacial-interglacial cycle is rather small. For example, Bock et al.,2017) write: "GEMs are in fact smaller than 47 (Holocene) and 41 (LGM) Tg CH4 a$-$1. " and "[GEM] are not strongly variable players

that could explain the observed glacial/interglacial [CH4] variations" If their analysis of their isotope measurements is correct, then the additional/anomalous source due to geological CH4 would only be 6 TgC/yr x 8,000 yr = 48 PgC over the past 8 ka. This is relatively small in comparison with the estimated peat accumulation of several hundred PgC.

In my opinion, it is appropriate for the explanation of CO2 variations to compare anomalous geological sources and sinks, representing deviations from the mean geological emissions (volcanoes, CH4, weathering) and mean geological sinks (sediment burial). Highlighting the magnitude of a selected individual flux such as total geological CH4 emissions appears misleading. It would be equally misleading to multiply the estimated weathering rate of ∼0.2-0.4 PgC yr with 8000 yr to get a NET source to the atmosphere of 1600 to 3200 PgC.

3) P9. L1: The simulated net atm-to-land fluxes may be compared with the observation-based reconstruction of Elsig et al., 2009

4) P9, l21: "On the other hand, simulations with intermediate complexity models suggested that the impact of the memory effect on Holocene carbon dynamics is rather minor (e.g., Menviel and Joos, 2012)".

This statement is not true. Please see table 4 in Menviel and Joos, 2012 for the 20 ppm CO2 increase over the last 7 ka: They attribute 10 ppm to legacy effects associated with land uptake during the transition and the early Holocene and 5 ppm to ocean-sediment interactions and only 5 ppm to coral reef buildup. Their attribution is in line with ice core CO2 and d13C and to some extent with reconstructed CO3–. Uncertainties exist and their estimate for the coral growth is based on Vecsei and Berger which represents a low estimate.

The legacy or memory effects cannot be easily dismissed as small. For example, a substantial early Holocene carbon uptake is implied by both the d13C record (Elsig et al, 2009) as well as by reconstructions of retreating ice sheets. Such an uptake

has consequences for CaCO3 compensation within the ocean and thus for the late Holocene CO2 increase. The authors may wish to revise the discussion on this topic.

5) The cumulative shallow water CaCO3 deposition over the last 8 kyr is at the high end of available estimates. This may be discussed in the manuscript (see also point 7 below)

FURTHER COMMENTS

1) Section 2, Could you please provide some additional information on the ocean sediment model. It would be illustrative to add a table showing the global fluxes (CaCO3, Alk, POM, opal, nutrients..) to the sediment in comparison with observational estimates.

2) Table 1: Additional explanation may be needed to understand table 1.

It would be great if a sign convention is selected that allows to simply add up all the numbers to get to a ~0 overall budget.

This illustrative table provides the cumulative C changes in PgC in the atm., ocean, land, and sediments. It also provides the weathering input, but I miss the corresponding flux from sediments to the lithosphere. I guess this is included in the sediments? Is the CaCO3 removed at the surface added to the sediment pool? I guess this is not the case?

The budget seems not to add up exactly. For simulation TRAF, I get 2303 PgC in the atm-land-ocean-sediment versus 2137+152= 2289 PgC; are this minor 14 Pg difference to to rounding/integration? How do I need to add the numbers for TRAF and TRAFc to get a closed budget?

Additionally, it may be illustrative to show anomalies in ocean-to-sediment flux for CaCO3 and POC in Table 5. It is not clear whether the POC flux to sediment remained roughly constant or not.

3) Figure 3 and P6, line 8/9 "Carbon sedimentation is high in upwelling zones, mainly in coastal areas and the tropical Pacific, and that causes strong accumulation patterns." It may be instructive to show fluxes to the sediment as diagnosed at the end of the spin up and anomalies relative to this initial flux. It is my interpretation - I may be wrong - that figure 3 shows changes in DIC plus the integrated ocean-to-sediment flux. If this is true, this may be a bit misleading as the change in DIC reflects a real change in store, while the accumulated transfer to sediment may be too a large extent balanced by wheathering; then the actual change in sediment/lithosphere is smaller. If my interpretation is wrong, then the spatial gradients in the ocean sink/source shown in Fig. 3 may be better explained.

4) P6, l26: "Natural changes in vegetation and tree cover are most pronounced for the time slice around 1 CE" Do you mean there are large changes at 1 CE or rather 1CE minus 6 kyr BCE?

5) P7, l11: "The simulated increase in land carbon storage before 2000 CE and de-crease afterwards is consistent with the changes in atmospheric d13CO2 (Schmitt et al., 2012)." The original Holcene data are given in Elsig et al, 2009 and the outcome should be compared with the reconstructed air-land fluxes presented by Elsig et al.

6) P 7, l19: "The global $CaCO_3$ export from surface to aphotic layer increases by about 5% between 6000 and 2000 BCE in both TRAF and TRAFc simulations and returns to the 6000 BCE level by the end of the simulation." Could you please provide an explanation for this change in $CaCO_3$ export. Is POM export also declining or is the rain ratio changing in response to changes in surface $CO_2/CO_3-$?

7) P7, l23: "Accounting for 7850 years of experimental length, the required excessive carbonate sedimentation in the shallow waters would be 3 Tmol/yr or at the lower bound of estimates of 3.35 to 12 Tmol/yr $CaCO_3$ accumulation proposed by Vecsei and Berger (2004) and Opdyke and Walker (1992)."

I am confused here. Dividing the 1224 PgC of excessive $CaCO_3$ deposition given in
Tab 1 by 7850 yr and by 12g/mol, yields 12 Tmol/yr and not 3 Tmol/yr. I think it would be illustrative to compare the cumulative surface CaCO3 removal of 1224 PgC with the cumulative estimates given by Vecsei and Berger for the last 8 ka. This number is likely around 300 PgC; (Vecsei and Berger suggest a cumulative deposition of 378 PgC over the last 14 kyr).

8) P8, line 10: Table 2 seems missing in the MS?

---

## Short Comment (SC1) · 2 May 2019

This paper (*Brovkin et al.*, 2019) uses atmospheric greenhouse gases (GHG: $CO_2$, $CH_4$, $N_2O$, plotted in Figure 1b,c of the discussion paper) from spline routines based on various data sets. Since such a GHG data compilation excercise including the calculation of a spline has also been performed in a recent study (*Köhler et al.*, 2017), I asked the corresponding author to get access to their applied GHG time series to evaluate if and how they might differ from the final splines of this other study. I plot them here against these earlier results in the following figures 1–3. Spline routines applied here and there have been the same (developed by Fortunat Joos, Universitiy of Bern), but the underlying data and the chosen prescribed cutoff period $P_c$ for the spline routines have been in detail slightly different leading to similar, but not identical

splines.

For $CO_2$ (Fig. 1) both splines are nearly identical.

The $CH_4$ (Fig. 2) record in *Köhler et al.* (2017) is based on the WAIS Divide Ice Core (WDC) for large parts of the Holocene, that resolves multi-cenntennial variabilies, a small-scale featue that is ignored in the spline used in *Brovkin et al.* (2019). This comparision also highlights, that the $CH_4$ data used in *Brovkin et al.* (2019) are not global mean values, but southern hemispheric values. Due to an existing interhemispheric gradient, northern hemispheric $CH_4$ (e.g. from Greenland ice cores) and therefore also global mean $CH_4$ values are slightly larger than the $CH_4$ values of the chosen southern hemispheric spline.

In $N_2O$ (Fig. 3) the millennial-scale variability is slightly shifted in time between both splines, suggesting that the used age modesl of the underlying data might have been different.

The spline used in *Brovkin et al.* (2019) fall nearly always into the uncertainty bands ($\pm 2\sigma$) of the splines described in *Köhler et al.* (2017).

For details of the spline method and further citations of the underlying data the reader is refered to *Köhler et al.* (2017). Layout of figures and captions have been adapted from the previous paper.

I believe these underlying details of the method and data might been of interest to the readers of *Brovkin et al.* (2019)

**References**

Brovkin, V., S. Lorenz, T. Raddatz, T. Ilyina, I. Stemmler, M. Toohey, and M. Claussen (2019), What was the source of the atmospheric $CO_2$ increase during the Holocene?, *Biogeosciences Discussions*, *2019*, 1–25, doi:10.5194/bg-2019-64.

Köhler, P., C. Nehrbass-Ahles, J. Schmitt, T. F. Stocker, and H. Fischer (2017), A 156 kyr smoothed history of the atmospheric greenhouse gases $CO_2$, $CH_4$, and $N_2O$ and their radiative forcing, *Earth System Science Data*, *9*, 363–387, doi:10.5194/essd-9-363-2017.

**Figure Captions**

**Figure 1:** Atmospheric $CO_2$ spline and underlying data (2016 CE – 8,000 BP). Black spline as published in *Köhler et al.* (2017) against time series (gold) used in *Brovkin et al.* (2019). Error bars around the ice core data points are $\pm 2\sigma$. WDC data have been adjusted to reduce offsets, see *Köhler et al.* (2017) for details.

**Figure 2:** Atmospheric $CH_4$ spline and underlying data (2016 CE – 8,000 BP). Black spline as published in *Köhler et al.* (2017) against time series (gold) used in *Brovkin et al.* (2019). Details on plotted data are explained in *Köhler et al.* (2017). The maximum ice core data uncertainty ($\pm 2\sigma$) is sketched in the lower left corner. Latitudinal origin of data is indicated by NH and SH, implying northern and southern hemisphere, respectively.

**Figure 3:** Atmospheric $N_2O$ spline and underlying data (2016 CE – 8,000 BP). Black spline as published in *Köhler et al.* (2017) against time series (gold) used in in *Brovkin et al.* (2019). Details on plotted data are explained in in *Köhler et al.* (2017). The maximum ice core data uncertainty ($\pm 2\sigma$) is sketched in the upper right corner. Filled symbols: data taken for spline; open symbols: data not taken for spline.

[Figure]

[Figure]

Fig. 1. Figure caption is contained at the end of text.

[Figure]

**Fig. 2.** Figure caption is contained at the end of text.

[Figure]

Fig. 3. Figure caption is contained at the end of text.

---

## Author Comment (AC1) · 10 May 2019

**Response to the Interactive comment by Anonymous Referee #1 (given in *italic* below)**

Brovkin et al. present the results of two transient simulations covering the period 6000 BCE to pre-industrial 1850 CE with the Earth system model MPI-ESM-LR. The goal is to constrain the processes leading to the changes in atmospheric CO2 concentration during that period. The conclusion of the study is that a surface alkalinity decrease, for example due to enhanced carbonate accumulation on continental shelves, is necessary to explain the Holocene atmospheric CO2 trajectory.

This is an interesting and very valuable study.

*We thank the reviewer for the positive evaluation of our study.*

Please find some comments below that should be addressed before publication.

1) Experimental set up:
The model was equilibrated under 6000 BCE conditions, with an atmospheric CO2 concentration lower (260 ppm) than during the pre-industrial. If the global alkalinity concentration was kept constant, then the ocean would have lost carbon during the spin-up phase. Then forced with a transient increase in atmospheric CO2, all else being equal, the ocean should take up carbon. This is apparently what is happening as changes in SST, ocean circulation. . . are small. Potential bias due to imposing an atmospheric CO2 concentration should be briefly mentioned.

*The global alkalinity in the spin-up experiments was not constant, as the carbonate chemistry was slowly responded to new boundary conditions. In the spinup experiment 8KAF (more than 1,000 years), the weathering flux has not been changed comparing to pre-industrial conditions. The change in the boundary conditions led to different sedimentation fluxes, which resulted in the slow decline of alkalinity in this experiment.*

*In the TRAFc experiment, which has interactive carbon cycle the weathering fluxes and surface alkalinity were adjusted to changes in the boundary conditions. The equilibration procedure in HAMOCC followed the same rules as in CMIP5 spin-up experiments (Ilyina et al., JAMES, 2013): "Throughout the equilibration process, weathering fluxes and $CaCO_3$ content in sediments have been changed, which led to changes in total alkalinity (TA). This would have occurred naturally, without leading to excess TA, had the biogeochemistry model been given a long-enough spin-up time to equilibrate its sediments. Along with change in TA, also DIC changed (with the molar ratio 2:1) in order to maintain the correct $pCO_2$." This equilibration procedure was applied in the spinup of the TRAFc experiment. In particular, first the system was switched to interactive carbon mode using the same weathering rates as in 8KAF for ~300 years, then it was stabilized the system by increase all of the weathering fluxes (Si, OM, $CaCO_3$) which led to a stabilization of the surface alkalinity, afterwards the alkalinity was increased to keep the target $pCO_2$. For the last few hundred years, a weathering was adjusted that lead to alkalinity stabilization. In total, equilibration spinup 8KAFc for the TRAFc experiment took more than 1,000 model years.*

*We will discuss different equilibration procedure and implications for interpretation of model results in the revised manuscript. In particular, if the TRAF simulation would start from the same initial conditions as the TRAFc, the ocean would not be a source of carbon from the 8*

*kyr BP, and at the end the ocean carbon uptake and mismatch of carbon budget would be higher than 166 PgC obtained in the TRAF experiment.*

2) Justification of surface alkalinity decrease and comparison with previous studies:
The authors have previously done extensive work on the topic of global carbon cycle changes on glacial-interglacial cycles. They are therefore well aware of the literature on the topic, on the rationale behind decreasing surface alkalinity during the Holocene, and on the results from previous studies. However, given that here it is given as the main mechanism controlling Holocene atmospheric CO2, I would have expected a more in depth introduction of the topic and discussion with respect to previous studies. There is no mention in the introduction of the timing and magnitude of carbonate sedimentation on shelves, as well as on the results of previous modelling studies on the topic. There are only a few words on the topic in the introduction (p2, L.13-14), a few words in the method without any quantification (p4, L.17-18). As a side note, the introduction given in Kleinen, Brovkin et al., (2016) was much more informative. The results and a rapid comparison to Vecsei and Berger (2004) is given p7, L 23-25, but there is no comparison with results from previous studies. In addition, the magnitude of the necessary alkalinity change and its equivalent change in carbonate sedimentation could also be discussed in the context of the simulated changes in land carbon (see section 3).

*Introduction: It is always difficult to decide how much of previously written overviews should be repeated in the next paper. As we extensively wrote on state-of-the-art of the Holocene carbon dynamics in the recent papers by Kleinen et al, Brovkin et al., (2016), we limited current introduction to discussion of key old papers (which choice is, of course, a bit subjective) and novel studies appeared after the recent reviews. One of original papers we definitely missed is the study by Elsig et al. (2009). We will refer to it in the revised manuscript, including comparison of their and our results. In particular, from the Fig. 3 in Elsig et al., one can conclude that the deconvolution approach resulted in the land uptake of ca. 140 PgC from 8 to 5 kyr BP, divided rather equally into ca. 70 PgC from 8 to 7 BP and 70 PgC from 7 to 5 kyr BP. The land $CO_2$ uptake from 8 to 7 BP deduced from the increase in atmospheric $\delta^{13}C$ by Elsig et al. could not be reproduced in our experiments because of equilibrium model assumptions at 8 kyr BP. Such a non-equilibrium response can be captured only in transient simulations during the last deglaciation. In our TRAF and TRAFc experiments, land accumulates about 60 PgC between 7 and 4 kyr BP. One can conclude that from 7 kyr BP, the land carbon dynamics (uptake of 60 PgC by 4 kyr BP, release of 80-100 PgC by 1850 mainly due to landuse) is similar to the land carbon changes provided by Eslig et al.*

3) Changes in terrestrial carbon:
Using a mass balance approach and high-resolution atmospheric CO2 and d13CO2 records, Elsig et al., 2009 suggest a land carbon uptake of 290 GtC between 11ka and 5 ka B.P. I am surprised to see no reference/discussion to this study. As far as I can see this result seems supported among others by Stocker et al., (2017), Menviel & Joos (2012) ... Here, the model suggests a terrestrial carbon uptake of 50 GtC between 8 and 2 ka B.P, which is much smaller and with a different timing. A discussion should be added on the results of this study, compared to the estimates of Elsig et al., (2009). The authors should discuss how their results could be reconciled with the atmospheric d13CO2 record, as also shown in other modelling studies, which included carbon isotopes.

*See our response above. We will add comparison with results by Elsig et al. (2009) into the revised manuscript.*

---

## Author Comment (AC3) · 10 May 2019

We thank Peter Koehler for this insightful figure and will refer to the comment in the revised manuscript

---

## Author Response (AR1)

**Response to the Interactive comment by Anonymous Referee #1 (given in *italic* below)**

Brovkin et al. present the results of two transient simulations covering the period 6000 BCE to pre-industrial 1850 CE with the Earth system model MPI-ESM-LR. The goal is to constrain the processes leading to the changes in atmospheric CO2 concentration during that period. The conclusion of the study is that a surface alkalinity decrease, for example due to enhanced carbonate accumulation on continental shelves, is necessary to explain the Holocene atmospheric CO2 trajectory.

This is an interesting and very valuable study.

*We thank the reviewer for the positive evaluation of our study.*

Please find some comments below that should be addressed before publication.

1) Experimental set up:
The model was equilibrated under 6000 BCE conditions, with an atmospheric CO2 concentration lower (260 ppm) than during the pre-industrial. If the global alkalinity concentration was kept constant, then the ocean would have lost carbon during the spin-up phase. Then forced with a transient increase in atmospheric CO2, all else being equal, the ocean should take up carbon. This is apparently what is happening as changes in SST, ocean circulation. . . are small. Potential bias due to imposing an atmospheric CO2 concentration should be briefly mentioned.

*The global alkalinity in the spin-up experiments was not constant, as the carbonate chemistry was slowly responded to new boundary conditions. In the spinup experiment 8KAF (more than 1,000 years), the weathering flux has not been changed comparing to pre-industrial conditions. The change in the boundary conditions led to different sedimentation fluxes, which resulted in the slow decline of alkalinity in this experiment. We discuss this equilibration procedure and implications for interpretation of model results in the revised manuscript (p. 3, l.32 - p.4, l. 11):*
*The weathering flux has not been changed comparing to pre-industrial conditions. The change in the boundary conditions to 6000 BCE led to slightly different sedimentation fluxes, which resulted in a slow decline of alkalinity in 8KAF. Afterwards, the model was run with an interactive carbon cycle to ensure a dynamic equilibrium between land, ocean, and atmospheric carbon cycle components (simulation 8KAFc). In the 8KAFc simulation, the equilibration procedure in HAMOCC followed the CMIP5 spinup procedure (Ilyina et al., 2013): "Throughout the equilibration process, weathering fluxes and $CaCO_3$ content in sediments have been changed, which led to changes in total alkalinity (TA). This would have occurred naturally, without leading to excess TA, had the biogeochemistry model been given a long-enough spin-up time to equilibrate its sediments. Along with change in TA, also DIC changed (with the molar ratio 2:1) in order to maintain the correct $pCO_2$." In the interactive carbon spinup, the model firstly used the same weathering rates as in 8KAF for ~300 years, then it stabilized the system by increase all of the weathering fluxes (Si, OM, $CaCO_3$) which led to a stabilization of the surface alkalinity, afterwards the alkalinity was changing to keep the target $pCO_2$. For the last few hundred years, a weathering was adjusted that lead to alkalinity stabilization. In total, the 8KAFc spinup took more than 1,000 model years.*

*We also added on the implications for interpretation of model results in the revised manuscript (p. 8, l. 17-20): "In particular, this alkalinity drift in TRAF explains the initial decrease in the ocean carbon storage until ca. 4500 BCE (Fig. 2a), despite of an increase in atmospheric CO₂ concentration. If TRAF would have started from an equilibrated system as TRAFc, the beginning of TRAF would have been more similar to TRAFc, and the ocean carbon uptake would have started earlier."*

2) Justification of surface alkalinity decrease and comparison with previous studies:
The authors have previously done extensive work on the topic of global carbon cycle changes on glacial-interglacial cycles. They are therefore well aware of the literature on the topic, on the rationale behind decreasing surface alkalinity during the Holocene, and on the results from previous studies. However, given that here it is given as the main mechanism controlling Holocene atmospheric CO2, I would have expected a more in depth introduction of the topic and discussion with respect to previous studies. There is no mention in the introduction of the timing and magnitude of carbonate sedimentation on shelves, as well as on the results of previous modelling studies on the topic. There are only a few words on the topic in the introduction (p2, L.13-14), a few words in the method without any quantification (p4, L.17-18). As a side note, the introduction given in Kleinen, Brovkin et al., (2016) was much more informative. The results and a rapid comparison to Vecsei and Berger (2004) is given p7, L 23-25, but there is no comparison with results from previous studies. In addition, the magnitude of the necessary alkalinity change and its equivalent change in carbonate sedimentation could also be discussed in the context of the simulated changes in land carbon (see section 3).

*Introduction: It is always difficult to decide how much of previously written overviews should be repeated in the next paper. As we extensively wrote on state-of-the-art of the Holocene carbon dynamics in the recent papers by Kleinen et al, Brovkin et al., (2016), we limited current introduction to discussion of key old papers (which choice is, of course, a bit subjective) and novel studies appeared after the recent reviews. One of original papers we definitely missed is the study by Elsig et al. (2009). We refer to it in the revised manuscript, including comparison of their and our results:*
*(p.2, l. 15-17)*
*Using deconvolution approach based on ice core CO₂ and δ¹³C data, Elsig et al. (2009) concluded on a significant fraction of Holocene CO₂ changes attributed to the carbonate compensation effects during deglaciation.*
*(p.7, l.22-30):*
*"The deconvolution approach (Fig. 3 in Elsig et al. (2009)) resulted in the land uptake of ca. 140 PgC from 6000 to 3000 BCE, divided rather equally into ca. 70 PgC from 6000 to 5000 BCE and 70 PgC between 5000 and 3000 BCE. The 70 PgC uptake from 6000 to 5000 BCE deduced from the increase in atmospheric δ¹³CO₂ is not reproduced in our experiments, likely because it is a non-equilibrium land response which can be captured only in transient simulations during the last deglaciation. In our TRAF and TRAFc experiments, land accumulates about 60 PgC between 5000 and 2000 BCE, comparable with land uptake of 70 PgC between 5000 and 3000 BCE inferred by Elsig et al. (2009). We can conclude that after 5000 BCE, the land carbon dynamics in MPI-ESM (uptake of 60 PgC by 2000 BCE, release of 80-100 PgC by 1850 CE, predominantly due to landuse) is similar to the land carbon changes estimated by Elsig et al. (2009)."*
*p.11,l.9-11:*
*This is in line with previous simulations performed with intermediate complexity models (e.g., Kaplan et al., 2002; Kleinen et al., 2016)* **and with ice core deconvolution studies (Elsig et al., 2009; Schmitt et al., 2012).**

3) Changes in terrestrial carbon:
Using a mass balance approach and high-resolution atmospheric $CO_2$ and d13CO2 records, Elsig et al., 2009 suggest a land carbon uptake of 290 GtC between 11ka and 5 ka B.P. I am surprised to see no reference/discussion to this study. As far as I can see this result seems supported among others by Stocker et al., (2017), Menviel & Joos (2012) ... Here, the model suggests a terrestrial carbon uptake of 50 GtC between 8 and 2 ka B.P, which is much smaller and with a different timing. A discussion should be added on the results of this study, compared to the estimates of Elsig et al., (2009). The authors should discuss how their results could be reconciled with the atmospheric d13CO2 record, as also shown in other modelling studies, which included carbon isotopes.

*See our response above. We added comparison with results by Elsig et al. (2009) into the revised manuscript (p.7, l.22-30).*

**Response to the Interactive comment by Fortunat Joos (given in *italic* below)**

The study by Victor Brovkin and colleagues is interesting. They provide results from first transient fully coupled ESM simulations covering the entire last 8000 years. This is novel and warrants publication.

*We thank Fortunat Joos for his helpful and constructive comments. Indeed, transient coupled climate-carbon cycle simulation with full-scale ESM is a challenge; our runs were taking half-a-year each.*

The conclusion by Brovkin et al. that shallow-water $CaCO3$ deposition (coral reef growth) plays a role for the late Holocene $CO_2$ increase is similar to the conclusions from earlier studies using EMICs. A difference is that this study seems to imply that shallow water carbonate deposition is by far the most important driver for the late Holocene $CO_2$ increase. This is a possibility, but others found additional factors such as legacy effects of earlier land carbon uptake to be equal or even more important.

*We agree that the other factors, mainly legacy/memory of the deglaciation period, are relevant too. In our study, we are of course limited by (i) equilibrium assumptions of spinup and (ii) land and ocean biogeochemistry parameterizations in our model. All what we can say is that implications of excessive shallow water carbonate deposition do not violate limitations of proxies (carbonate sedimentation, carbonate ion changes).*

Here below my specific comments in addition to those offered by reviewer 1.

1. Information about model drift may be helpful for the reader.

*We cannot say much on the ocean biogeochemistry drift in the TRAFc experiment as there is no control experiment (without forcings) for 8,000 years, and making new simulation is not possible. The TRAF experiment was less properly initialized, as the weathering was not adjusted to changes in boundary conditions (see our response to the reviewer #1 comments),*

*causing substantial part of surface alkalinity decreases in the TRAF experiment (Fig. 6c). This mismatch explains the initial decrease in the ocean carbon storage until 6.5 ka BP (Fig. 2a), despite of an increase in atmospheric $CO_2$ concentration. If TRAF would have started from an equilibrated system as TRAFc, the beginning of TRAF would have been more similar to TRAFc (close to no net co2 flux) and the uptake would have started earlier.*
*We discuss this on p.8, l.15-20:*
*Let us note that in the 8KAF and TRAF experiments the weathering was not adjusted to changes in boundary conditions and this likely caused surface alkalinity decrease in the transient run (Fig. 7c). In particular, this alkalinity drift in TRAF explains the initial decrease in the ocean carbon storage until ca. 4500 BCE (Fig. 2a), despite of an increase in atmospheric $CO_2$ concentration. If TRAF would have started from an equilibrated system as TRAFc, the beginning of TRAF would have been more similar to TRAFc, and the ocean carbon uptake would have started earlier.*
*And p.10, l. 16-20;*
*In particular, as POC fluxes to the sediment are not properly compensated by the fixed weathering, this leads to changes in nutrient inventory in transient simulations. Both, $CaCO_3$ and POC fluxes to sediments are changing with time; this also leads to changes in the rain ratio. In the absence of factorial experiments without these changes, it is difficult to infer how do these trends in nutrients and biogenic opal and carbonate fluxes affect atmospheric $CO_2$. These two caveats (steady-state initial conditions and fixed weathering) apply to both TRAF and TRAFc simulations.*

P4, l25-l29: I am puzzled about the large, 50%, difference in the diagnosed weathering flux between the simulations TRAF and TRAFc.

*This difference is because during the spinup phase of TRAFc (as discussed in response to the referee #1) weathering rates were adjusted to match the net loss to the sediment, whereas this was not done for the spinup run of TRAF. Over 7850 years, this is large difference (2137 vs 3270 PgC in TRAF and TRAFc, respectively). On the other hand, large weathering in TRAFc is compensated by immediate surface $CaCO_3$ removal in this experiment (1224 PgC). Total alkalinity in TRAFc is affected by 3270-1224=2046 PgC weathering, which is comparable with 2137 PgC weathering in the TRAF experiment, so that as a result net weathering in TRAFc is less than in TRAF leading to decrease in the total alkalinity. We added a new figure 3 with which explains the total carbon budget in both experiments.*

P3, l30: Is the ALK nudging also used during the coupled spin up 8KAFc. If not, how large is the drift in CO2? Both TRAF and TRAFc were first spin up under prescribed CO2 (260 ppm, 8KAF). The spin up is extended by an additional 100 years with an open atmosphere (simulation 8KAFc) before starting TRAFc. The weathering flux is diagnosed from the last 300 yr of the spin up. In other words, the last 200 years of 8KAF are used to diagnose the weathering for TRAF and TRAFc; the difference in the diagnosed weathering for TRAF and TRAFcarises from the other 100 years of results taken either from 8KAF or from 8KAFc. Why is there such a large difference in the diagnosed weathering flux even though 200 out of 300 years are taken from the same run? Is the model far from equilibrium? Is there a substantial model drift? Is there information from a control run available?

*The alkalinity was not nudged in the spin up 8KAFc, but during the spinup procedure the alkalinity and weathering fluxes were adjusted as explained above. During the last several hundred years, the alkalinity was stable. The CO2 was fluctuating responding to the climate/weather variability was changing, but the drift in the CO2 during the last 100 years of spinup was negligible. Over such a long run, the constant weathering cannot perfectly*

*compensate for a nonlinear evolution of the sediment. Expalanation of weathering spinup was added to Methods, p.3 l. 32- p.4, l.12:*

*The spinup simulation 8KAF started from initial conditions for pre-industrial climate and continued with boundary conditions for 6000 BCE for more than 1,000 years in order to establish an equilibrium of climate and carbon cycle with the boundary conditions. During the spinup period the atmospheric $CO_2$ concentration was kept at a constant level of 260 ppm. The weathering flux has not been changed comparing to pre-industrial conditions. The change in the boundary conditions to 6000 BCE led to slightly different sedimentation fluxes, which resulted in a slow decline of alkalinity in 8KAF. Afterwards, the model was run with an interactive carbon cycle to ensure a dynamic equilibrium between land, ocean, and atmospheric carbon cycle components (simulation 8KAFc). In the 8KAFc simulation, the equilibration procedure in HAMOCC followed the CMIP5 spinup procedure (Ilyina et al., 2013): "Throughout the equilibration process, weathering fluxes and $CaCO_3$ content in sediments have been changed, which led to changes in total alkalinity (TA). This would have occurred naturally, without leading to excess TA, had the biogeochemistry model been given a long-enough spin-up time to equilibrate its sediments. Along with change in TA, also DIC changed (with the molar ratio 2:1) in order to maintain the correct $pCO_2$." In the interactive carbon spinup, the model firstly used the same weathering rates as in 8KAF for ~300 years, then it stabilized the system by increase all of the weathering fluxes (Si, OM, $CaCO_3$) which led to a stabilization of the surface alkalinity, afterwards the alkalinity was changing to keep the target $pCO_2$. For the last few hundred years, a weathering was adjusted that lead to alkalinity stabilization. In total, the 8KAFc spinup took more than 1,000 model years.*

*Also, we discuss consequences of fixed weathering for nutrients at the end of the discussion, p.10, 1.10-16:*
*"In particular, as POC fluxes to the sediment are not properly compensated by the fixed weathering, this leads to changes in nutrient inventory in transient simulations. Both, $CaCO_3$ and POC fluxes to sediments are changing with time; this also leads to changes in the rain ratio. In the absence of factorial experiments without these changes, it is difficult to infer how do these trends in nutrients and biogenic opal and carbonate fluxes affect atmospheric $CO_2$. These two caveats (steady-state initial conditions and fixed weathering) apply to both TRAF and TRAFc simulations."*

2) The statement on geological methane emissions appears misleading and needs to be revised.

P8, line 28: "Geological sources of methane of the scale of 30-40 Tg/yr are pronounced in intergacials (Bock et al., 2017; Saunois et al., 2016). Although uncertainty in the geological methane source remains high, after oxidation in the atmosphere, this source would correspond to 200-300 GtC during the last 8,000 years and potentially compensate for a substantial part of the peat growth." The change in geological methane emissions (GEM) over glacial-interglacial cycle is rather small. For example, Bock et al.,2017) write: "GEMs are in fact smaller than 47 (Holocene) and 41 (LGM) Tg $CH_4$ a–1. " and "[GEM] are not strongly variable players that could explain the observed glacial/interglacial [$CH_4$] variations" If their analysis of their isotope measurements is correct, then the additional/anomalous source due to geological $CH_4$ would only be 6 TgC/yr x 8,000 yr = 48 PgC over the past 8 ka. This is relatively small in comparison with the estimated peat accumulation of several hundred PgC.

In my opinion, it is appropriate for the explanation of $CO_2$ variations to compare anomalous geological sources and sinks, representing deviations from the mean geological emissions (volcanoes, $CH_4$, weathering) and mean geological sinks (sediment burial). Highlighting the

magnitude of a selected individual flux such as total geological CH4 emissions appears misleading. It would be equally misleading to multiply the estimated weathering rate of _0.2-0.4 PgC yr with 8000 yr to get a NET source to the atmosphere of 1600 to 3200 PgC.

*Our initial point was that the geological CH$_4$ sources have a large uncertainty, and CO$_2$ flux from oxidized methane is missing in the coupled model setup, but we agree with the reviewer that it is misleading to compare with total sum and not the net effect. We take this argument out of revised paper, and added a point about potential carbon release from the permafrost region, p.9, l. 23-25:*
*"On the other hand, we neglect other sources of atmospheric CO2 which might at least partly compensate for the peatland growth, for example, emissions due to ongoing thermokarst formation and erosion of permafrost soils, especially close to the Arctic coast (Lindgren et al., 2018)."*
3) P9. L1: The simulated net atm-to-land fluxes may be compared with the observation-based reconstruction of Elsig et al., 2009

*We added this comparison to the revised manuscript, see our reply to the reviewer #1*

4) P9, l21: "On the other hand, simulations with intermediate complexity models suggested that the impact of the memory effect on Holocene carbon dynamics is rather minor (e.g., Menviel and Joos, 2012)".

This statement is not true. Please see table 4 in Menviel and Joos, 2012 for the 20 ppm CO2 increase over the last 7 ka: They attribute 10 ppm to legacy effects associated with land uptake during the transition and the early Holocene and 5 ppm to ocean-sediment interactions and only 5 ppm to coral reef buildup. Their attribution is in line with ice coreCO2 and d13C and to some extent with reconstructed CO3–. Uncertainties exist and their estimate for the coral growth is based on Vecsei and Berger which represents a low estimate.

The legacy or memory effects cannot be easily dismissed as small. For example, a substantial early Holocene carbon uptake is implied by both the d13C record (Elsig et al, 2009) as well as by reconstructions of retreating ice sheets. Such an uptake has consequences for CaCO3 compensation within the ocean and thus for the late Holocene CO2 increase. The authors may wish to revise the discussion on this topic.

*Apparently, Menviel and Joos (2009) wrote "… our results indicate that coral reef growth and other shallow-water carbonate deposition are a major contributor for the atmospheric CO2 rise after 7 ka B.P.", but indeed this doesn't mean that contributions due to other processes are minor. We revised the discussion according to the reviewer comments: p.10, l.8-10:*
*"Simulations with intermediate complexity models suggested that the impact of the memory effect from deglaciation on Holocene carbon dynamics, in particular due to carbonate compensation, is significant (e.g., Menviel and Joos, 2012)"*
5) The cumulative shallow water CaCO3 deposition over the last 8 kyr is at the high end of available estimates. This may be discussed in the manuscript (see also point 7 below)

*We agree that the cumulative shallow water CaCO$_3$ deposition as indicated in the table is confusing, as it is counterbalanced by much higher weathering, see our response to the weathering point above. We revised the text as follows, (p.8, l.10-20):*
*Accounting for 7850 years of experimental length, the alkalinity loss corresponds to 8.2 and 11.2 Tmol/yr CaCO$_3$ sedimentation in TRAF and TRAFc simulations, respectively. The*

*required excessive carbonate sedimentation in the shallow waters would be 3 Tmol/yr in TRAFc relative to TRAF, or at the lower bound of estimates of 3.35 to 12 Tmol/yr CaCO$_3$ accumulation proposed by Vecsei and Berger (2004) and Opdyke and Walker (1992). Even corresponding excessive carbonate sedimentation of 11.2 Tmol/yr CaCO$_3$ in the TRAFc simulation would fall into this observational range, although at the higher bound. Let us note that in the 8KAF and TRAF experiments the weathering was not adjusted to changes in boundary conditions and this likely caused surface alkalinity decrease in the transient run (Fig. 7c). In particular, this alkalinity drift in TRAF explains the initial decrease in the ocean carbon storage until ca. 4500 BCE (Fig. 2a), despite of an increase in atmospheric CO$_2$ concentration. If TRAF would have started from an equilibrated system as TRAFc, the beginning of TRAF would have been more similar to TRAFc, and the ocean carbon uptake would have started earlier."*

**FURTHER COMMENTS**

1) Section 2, Could you please provide some additional information on the ocean sediment model. It would be illustrative to add a table showing the global fluxes (CaCO3, Alk, POM, opal, nutrients..) to the sediment in comparison with observational estimates.

*We see the reviewer point, but think that it is more informative to compare sediment states rather than fluxes for several reasons. Particle fluxes to the sediment of POM, opal, or CaCO$_3$ reflect the respective production pattern and remineralisation/dissolution length scales (i.e. for calc the lysocline depth), thus the water column state, whereas there are no data to assess the global diffusive fluxes between sediment pore water and water column. For the CaCO$_3$ distribution, please see below the comparison of distribution in the piControl simulation of the current model version with observations. Generally, spatial state structures in our sediment model are similar to observations and driven by particle fluxes and their respective remineralization length scales in the water column (e.g. OM maxima in the tropical Pacific, opal maxima in the Southern ocean) and lysocline depth (and opal vs calc shell formation) for calcite. The general performance of the sediment model is described by Heinze et al. (1999). As an example, we provided a map comparison of CaCO$_3$ in coretop of marine sediments: data (top), model (bottom) in the interactive comment, and added the following sentence in the revised manuscript (p.6, l.20-22):*
*„These sedimentation patterns are typical for the HAMOCC model with interactive sediments (Heinze et al., 1999); they are generally well comparable with observed sedimentation patterns for organic carbon and CaCO$_3$ (Seiter et al., 2004; Archer, 1996)."*

2) Table 1: Additional explanation may be needed to understand table 1. It would be great if a sign convention is selected that allows to simply add up all the numbers to get to a _0 overall budget.
This illustrative table provides the cumulative C changes in PgC in the atm., ocean, land, and sediments. It also provides the weathering input, but I miss the corresponding flux from sediments to the lithosphere. I guess this is included in the sediments? Is the CaCO3 removed at the surface added to the sediment pool? I guess this is not the case?

*In MPI-ESM, lithosphere is not an explicit component of the carbon cycle. According to the HAMOCC sediment model (Heinze et al, 1999), the sediment layer flux from the active layer (12 cm) to deeper layers is effectively a flux to lithosphere as deeper layers cannot be*

*dissolved (constitute an ultimate loss). We do not report this flux to lithosphere explicitly. The CaCO3 removed at the surface is not added to the sediment pool. What is ultimately lost to the sediment is compensated for by the constant weathering fluxes.*

The budget seems not to add up exactly. For simulation TRAF, I get 2303 PgC in the atm-land-ocean-sediment versus 2137+152= 2289 PgC; are this minor 14 Pg difference to rounding/integration? How do I need to add the numbers for TRAF and TRAFc to get a closed budget?

*Indeed, simulation TRAF doesn't have a closed carbon budget, while it is closed for TFAFc. To illustrate how the budget needs to be calculated, we added a new Figure 3 to the revised manuscript.*

Additionally, it may be illustrative to show anomalies in ocean-to-sediment flux for CaCO3 and POC in Table 5. It is not clear whether the POC flux to sediment remained roughly constant or not.

*Indeed, the POC flux to sediments is not constant; as the weathering is constant, nutrient loss to the sediment is not properly compensated for the POC fluxes to the sediment.*
*We do not have the table 5 in the manuscript. We added a following explanation into discussion, p.10, l.16-20:*
*"In particular, as POC fluxes to the sediment are not properly compensated by the fixed weathering, this leads to changes in nutrient inventory in transient simulations. Both, $CaCO_3$ and POC fluxes to sediments are changing with time; this also leads to changes in the rain ratio. In the absence of factorial experiments without these changes, it is difficult to infer how do these trends in nutrients and biogenic opal and carbonate fluxes affect atmospheric $CO_2$."*

3) Figure 3 and P6, line 8/9 "Carbon sedimentation is high in upwelling zones, mainly in coastal areas and the tropical Pacific, and that causes strong accumulation patterns." It may be instructive to show fluxes to the sediment as diagnosed at the end of the spin up and anomalies relative to this initial flux. It is my interpretation - I may be wrong - that figure 3 shows changes in DIC plus the integrated ocean-to-sediment flux. If this is true, this may be a bit misleading as the change in DIC reflects a real change in store, while the accumulated transfer to sediment may be too a large extent balanced by weathering; then the actual change in sediment/lithosphere is smaller. If my interpretation is wrong, then the spatial gradients in the ocean sink/source shown in Fig. 3 may be better explained.

*We thank reviewer for this comment; indeed, we subtracted weathering then constructed this figure. We added this to the Figure caption.*

4) P6, l26: "Natural changes in vegetation and tree cover are most pronounced for the time slice around 1 CE" Do you mean there are large changes at 1 CE or rather 1CE minus 6 kyr BCE?

*We mean 1 CE minus 6 kyr BCE, clarify this in the paper, p.7, l. 5-6:*
"Natural changes in vegetation and tree cover are most pronounced for **the period before 1 CE**, before the start of substantial landuse forcing"

5) P7, l11: "The simulated increase in land carbon storage before 2000 CE and decrease afterwards is consistent with the changes in atmospheric d13CO2 (Schmitt et al., 2012)." The original Holocene data are given in Elsig et al, 2009 and the outcome should be compared with the reconstructed air-land fluxes presented by Elsig et al.

*We now compare with fluxes from Elsig et al. (2009), see response to the comments of the referee #1.*

6) P 7, l19: "The global CaCO3 export from surface to aphotic layer increases by about 5% between 6000 and 2000 BCE in both TRAF and TRAFc simulations and returns to the 6000 BCE level by the end of the simulation." Could you please provide an explanation for this change in CaCO3 export. Is POM export also declining or is the rain ratio changing in response to changes in surface CO2/CO3–?

*Both, CACO3 and POC fluxes to sediments are not constant; as the weathering is fixes, losses to the sediment are not properly compensated. This also leads to changes in the rain ratio. We discuss this in the revised paper, p.10:*
„Both, $CaCO_3$ and POC fluxes to sediments are changing with time; this also leads to changes in the rain ratio. In the absence of factorial experiments without these changes, it is difficult to infer how do these trends in nutrients and biogenic opal and carbonate fluxes affect atmospheric $CO_2$."

7) P7, l23: "Accounting for 7850 years of experimental length, the required excessive carbonate sedimentation in the shallow waters would be 3 Tmol/yr or at the lower bound of estimates of 3.35 to 12 Tmol/yr CaCO3 accumulation proposed by Vecsei and Berger (2004) and Opdyke and Walker (1992)."
I am confused here. Dividing the 1224 PgC of excessive CaCO3 deposition given in Tab 1 by 7850 yr and by 12g/mol, yields 12 Tmol/yr and not 3 Tmol/yr. I think it would be illustrative to compare the cumulative surface CaCO3 removal of 1224 PgC with the cumulative estimates given by Vecsei and Berger for the last 8 ka. This number is likely around 300 PgC; (Vecsei and Berger suggest a cumulative deposition of 378 PgC over the last 14 kyr).

*We agree that this could cause a confusion. Here, we meant a difference between TRAF and TRAFc experiments. It is caused by counteracting of CaCO3 removal by higher weathering flux at the surface; see a comment to the weathering point above. As seen in the Table 2 missing in the manuscript (see response to the point 8 below), additional CaCO3 removal accounting for extra weathering in TRAFc is much smaller, about 3 Tmol/yr.*
*We modified the paragraph (p.8, l.10-15) as follows:*
*„Accounting for 7850 years of experimental length, the alkalinity loss corresponds to 8.2 and 11.2 Tmol/yr $CaCO_3$ sedimentation in TRAF and TRAFc simulations, respectively. The required excessive carbonate sedimentation in the shallow waters would be 3 Tmol/yr in TRAFc relative to TRAF, or at the lower bound of estimates of 3.35 to 12 Tmol/yr $CaCO_3$ accumulation proposed by Vecsei and Berger (2004) and Opdyke and Walker (1992). Even corresponding excessive carbonate sedimentation of 11.2 Tmol/yr $CaCO_3$ in the TRAFc simulation would fall into this observational range, although at the higher bound."*

8) P8, line 10: Table 2 seems missing in the MS?

*We apologize as in the last-minute update, we forgot to add the table to the paper draft. We added the Table 2 into revised manuscript. It is useful for comparison of carbon fluxes between recent synthesis of data by Cartapanis et al (2018) and model experiments.*

**Response to the interactive comment by Peter Köhler:**
*We refer to his interactive comment in the revised paper (p.3, l.20):*
*" **see also comment by P. Köhler (2019)"***
*Köhler, P.: Interactive comment on "What was the source of the atmospheric $CO_2$ increase during the Holocene?"*
*by V. Brovkin et al., Biogeosciences Discussions, 10.5194/bg-2019-64-SC1, 2019.*

**Response to the comment by the editor**

It could be worthwhile to explain in a bit more detail the potential uncertainties associated with potential model drift issues.

*We discussed in more details the potential drift in alkalinity (p.8, l.15-20):*

[revised manuscript text omitted]

Figure 1

[Figure]

Fig. 2

[Figure]

Fig. 3

[Figure]

Fig. 4

[Figure]

Fig. 5

[Figure]

Fig. 6

[Figure]

Fig. 7

**TRAFc - TRAF**

[Figure]

Fig. 8